# Investigation of Indoor Air Quality in Residential Buildings by Measuring CO_2_ Concentration and a Questionnaire Survey

**DOI:** 10.3390/s22197331

**Published:** 2022-09-27

**Authors:** Shunichi Hattori, Toshiya Iwamatsu, Teruhisa Miura, Fujio Tsutsumi, Nobuyuki Tanaka

**Affiliations:** Central Research Institute of Electric Power Industry, 2-6-1 Nagasaka, Yokosuka-shi 240-0196, Kanagawa, Japan

**Keywords:** indoor air quality, ventilation, CO_2_ concentration, in-home sensing

## Abstract

Indoor air quality (IAQ) in houses is often deteriorated by chemical substances emitted from heating, building materials, or other household goods. Since it is difficult for occupants to recognize air pollution, they rarely understand the actual conditions of the IAQ. An investigation into the actual condition of IAQ in houses was therefore conducted in this study. Carbon dioxide (${\mathrm{CO}}_{2}$) concentrations in 24 occupied houses was measured, and the results from our analysis showed that the use of combustion heaters increased the concentration of ${\mathrm{CO}}_{2}$ and led to indoor air pollution. Results indicate that as outdoor temperature decreased, the frequency of ventilation decreased simultaneously, and ${\mathrm{CO}}_{2}$ concentration increased. Results of the questionnaire survey revealed that the actual IAQ in each house did not match the level of awareness its occupants had regarding ventilation. Along with this difficulty in perceiving air pollution, the lack of knowledge about ventilation systems and the effects of combustion heating may be additional barriers to IAQ awareness.

## 1. Introduction

This study investigates the actual indoor air quality (IAQ) in Japanese houses by measuring carbon dioxide (${\mathrm{CO}}_{2}$) concentrations and using questionnaire surveys.

Modern people spend most of their day indoors [1,2]. The IAQ in residential buildings is often poor, polluted by chemical substances emitted from heating appliances, building materials, or other household goods. Bolstered by the increase in highly airtight houses, there have been rising concerns about the increased health risks to occupants due to exposure to indoor pollutants such as ${\mathrm{CO}}_{2}$, carbon monoxide (CO), nitrogen oxides (${\mathrm{NO}}_{x}$), and volatile organic compounds (VOCs). Despite this concern, it has proven difficult for occupants to recognize air pollution, thus preventing them from understanding the actual IAQ conditions in their homes.

Ventilation is generally recommended to prevent IAQ deterioration. In Japan, the Building Standard Law was amended to require the installation of mechanical ventilation systems in all houses built after July 2003; however, this law only mandated the installation of ventilation equipment, and it is left to occupants to ensure that they are properly ventilating their homes. In addition, houses built before July 2003 were not considered for this mandatory ventilation installation. In Japan, roughly 26% of houses were estimated to have been built since 2003 relative to the total number of houses as of 2018 [3,4]. Multiple studies have reported that the amount of ventilation obtained by activities like opening windows decreases in the winter season [5,6], and there are concerns about the deterioration of air quality in highly airtight houses in recent years.

Another factor that causes air indoor pollution is the use of combustion heaters [7,8,9,10,11]. Combustion heating is recognized as a major source of ${\mathrm{CO}}_{2}$, sulfur dioxide (${\mathrm{SO}}_{2}$), and ${\mathrm{NO}}_{x}$ emissions [8,10,12], and exposure to these substances can cause sick building syndrome [13,14]. Gas and kerosene heaters still account for a large percentage of heating devices used in Japan [15,16,17], and the use of combustion heating has been reported to cause ${\mathrm{CO}}_{2}$ concentrations to reach 4000–5000 ppm [9,10,11].

The ${\mathrm{CO}}_{2}$ concentration is an indicator of the retention of harmful substances, as its concentration increases with inadequate ventilation in buildings [18,19,20,21,22]. Many countries have regulatory standards and guidelines to maintain ${\mathrm{CO}}_{2}$ concentrations below a fixed level (around 1000 to 1500 ppm) within buildings such as offices, schools, and houses [18,22,23]. The ${\mathrm{CO}}_{2}$ concentration also increases with human breathing; in addition to being an indicator of the retention of harmful substances, high concentrations are also known to increase the risk of respiratory diseases, impair thinking, and lower concentration [24,25,26,27]. It has also been reported that high concentrations of ${\mathrm{CO}}_{2}$ can affect the quality of sleep as well as overall performance the next day [28].

From the studies mentioned above, it can be gathered that there are two major factors that lead to air pollution in houses: inadequate ventilation and the use of combustion heaters. It is also worth considering the influence of outdoor temperature, because several studies suggest that seasonal changes affect ventilation rates [5,6,29,30]. Another important aspect is the difficulty for occupants to recognize the deterioration of IAQ. If occupants are not aware of the conditions in their immediate environment, it will be difficult for them to take appropriate actions to maintain a healthy IAQ.

Existing studies on the direct measurement of ventilation rate involve fan pressurization [5,31,32,33] and tracer gas [6,13,29,30], which are extremely intrusive and require expensive instruments and experts. Non-intrusive approaches that can be measured easily and at a low cost are required for widely adopted residential IAQ monitoring. ${\mathrm{CO}}_{2}$ concentration is therefore focused on because it is utilized as an indicator of IAQ and can be measured easily and at low cost.

In this study, a ${\mathrm{CO}}_{2}$-based investigation was conducted on the actual conditions of IAQ in Japanese houses, focusing on the following three aspects:(i)The relationship between combustion heating and indoor ${\mathrm{CO}}_{2}$ concentration.(ii)The influence of outdoor temperature on indoor ${\mathrm{CO}}_{2}$ concentration.(iii)The gap between occupants’ awareness of ventilation and the actual IAQ in their homes.

A ${\mathrm{CO}}_{2}$ sensor kit, which is inexpensive and easy to handle, was developed in this study. The ${\mathrm{CO}}_{2}$ concentration in houses was measured from autumn to winter, targeting houses with different housing attributes and heating methods. A questionnaire on occupants’ knowledge and awareness of indoor ventilation was also administered. Based on the results of these investigations, the influence of combustion heating, outdoor temperature, and occupants’ attitudes on the residential IAQ are discussed hereafter.

## 2. Background

### 2.1. ${\mathrm{CO}}_{2}$ Concentration as an Indicator of IAQ

${\mathrm{CO}}_{2}$ concentration is widely utilized as a comprehensive indicator of air pollution in IAQ [18,19,20,21,22]. Table 1 summarizes the ${\mathrm{CO}}_{2}$ concentration standards and guidelines for various countries [18,22,23]. In many countries, standards and guidelines are set at 1000–1500 ppm. In Canada, a 24 h average value is defined as the exposure time. Health Canada set its acceptable long-term exposure range to 3500 ppm in its 1987 report [34]; however, this standard was changed to 1000 ppm (24-h average) in 2021 [18]. In the United States, the standard is set at 700 ppm above the outdoor level [23]. Atmospheric ${\mathrm{CO}}_{2}$ concentrations have continued to rise year after year, with a global average of 414.72 ppm as of 2021 [35]. In Japan, the Building Standard Law sets the ${\mathrm{CO}}_{2}$ concentration standard at 1000 ppm for buildings within a certain area, while there are no standards or guidelines for residential buildings. In addition, no specific exposure times are determined in this law.

Indoors with high ${\mathrm{CO}}_{2}$ concentrations tend to retain harmful substances such as VOCs and ${\mathrm{PM}}_{2.5}$, because of inadequate ventilation. The second major source of pollutants in residential buildings is the use of combustion heating appliances (typically in winters). There is a close relationship between the use of combustion heating and air quality, especially the ${\mathrm{CO}}_{2}$, CO, ${\mathrm{NO}}_{x}$, ${\mathrm{SO}}_{2}$, and VOCs concentration, as reported in several pre-existing studies [8,9,10,11]. Combustion heating with gas or kerosene is still the main heating method in Japan [15,16,17], and the WHO reports that these remain in use in some parts of the world [7].

Exposure to these harmful substances, due to a lack of ventilation, has been reported to increase the risk of various respiratory diseases [36,37,38,39]. For example, Sun et al. investigated the relationship between the amount of ventilation and sick building syndrome in Chinese homes [13].

Simultaneously, high ${\mathrm{CO}}_{2}$ concentrations may have adverse effects of their own. As mentioned in Section 1, ${\mathrm{CO}}_{2}$ concentration increases with human breathing as well as the use of combustion appliances, and high ${\mathrm{CO}}_{2}$ concentrations are also known to increase the risk of respiratory diseases, impair thinking, and lower concentration, quality of sleep and overall performance the next day [24,25,26,27,28].

### 2.2. In-Home Sensing with IoT Sensors

Studies on residential monitoring have been increasing owing to the highly economical Internet of Things (IoT) sensors [40,41,42]. In-home sensing is mainly utilized for healthcare monitoring systems, which are collectively called *ambient assisted living* [43,44,45,46].

On-site real-time measurement of the ventilation rate can be replaced by the collection of ${\mathrm{CO}}_{2}$ sensor data. As mentioned in Section 1, direct ventilation measurement is both intrusive and expensive, and is also difficult to measure for a long period of time in a large number of houses. Sensor-based approaches are therefore expected to alleviate the impact on research participants, while achieving cost-effective measurements. These advantages allow ventilation monitoring for larger numbers of houses over a longer period of time.

Existing studies on IAQ in houses, described in Section 2.1, focus on the ${\mathrm{CO}}_{2}$ concentration as an air quality indicator and investigate its relationship with ventilation and combustion heating. However, comparisons among different heating appliances and the effects of seasonal variations are limited, and there are still unknowns in these areas. Existing studies [8,9,10,11] on combustion heating only investigated the impact of combustion, and did not compare it to homes without combustion heating (i.e., they had no controls in their study for comparison). Shinohara et al. observed the influence of the amount of ventilation in the house and seasonal variation, but argued that individual differences were much higher in Japan [6]. The following aspects are therefore necessary for a accurate investigation of the IAQ:

First, a survey must be conducted on the various patterns of indoor heating, in addition to houses using combustion heating. It is necessary to investigate air quality in a wide range of homes, including those that use only combustion heating, those that do not use combustion heating, and those that use multiple heating systems; this will clarify the relationship with the heating appliances used.

Second, the aforementioned investigation should not be limited to a short period of time, but should be conducted to consider seasonal changes such as in autumn and winter. There is a close relationship between awareness of ventilation and IAQ, which is greatly affected by outdoor temperature. For example, measuring the transition from autumn to winter would enable an investigation that is closer to the actual situation.

## 3. Experimental Design

### 3.1. Scope of the Investigation

The ${\mathrm{CO}}_{2}$ concentration is investigated as an indicator closely related to IAQ, and actual measurement of the ${\mathrm{CO}}_{2}$ concentration in houses was conducted for six months, i.e., from 1 October 2020, to 31 March 2021. Results of the questionnaire survey on the awareness of ventilation and the implementation of ventilation in each season are also considered, and their relationship with the measured IAQ is discussed.

As mentioned in Section 1, in order to clarify the actual state of the IAQ in houses, it is necessary to conduct an investigation focusing on three points: (1) the influence of heating appliances on IAQ; (2) the influence of outdoor temperature; and (3) the gap between occupants’ awareness of ventilation and actual IAQ. In Section 4, experimental results of the measurement of the ${\mathrm{CO}}_{2}$ concentration are analyzed from these points of view. The results for (1) and (2) are described in Section 4.1, focusing on the measurement results, and results for (3) are mentioned in Section 4.2, focusing on results of the questionnaire.

### 3.2. Research Participants

In order to measure the ${\mathrm{CO}}_{2}$ concentrations in occupied homes, 24 research participants living in urban areas were selected; Table 2 lists the participants’ houses and housing attributes. As shown in Figure 1, to exclude extremely warm or cold regions, participants’ houses were selected from the same climatic zone as Tokyo, according to definitions by Japan’s Ministry of Land, Infrastructure, Transport and Tourism [47]; Table 3 lists the climate data for Tokyo.

Participants’ houses were selected to have a diverse, holistic representation of types of dwelling, their structure, and the age of the building. Since the purpose of this experiment was to investigate the relationship between the heating equipment used and air pollution, we selected 14 houses that do not use combustion heating equipment and 10 houses that do. As shown in Table 2, most houses that use combustion heating also use multiple heating appliances such as air conditioners (ACs).

### 3.3. Measurements and Questionnaires

Experiments consisted of measurements in the participants’ houses and a questionnaire survey completed by the participants.

A sensor kit (see Figure 2) was installed in the living room of each house to measure the indoor ${\mathrm{CO}}_{2}$ concentration and temperature. This kit was developed based on the Raspberry Pi 3 B+, and measurement data were automatically uploaded to a cloud server through an LTE modem. ${\mathrm{CO}}_{2}$ concentration was measured using a Figaro CDM7160-C00 gas sensor, which can measure in the range of 360–5000 ppm in 1 ppm increments, with an accuracy of ±(50 ppm + 3%), and is pre-calibrated. An OMRON 2JCIE-BU01 environmental sensor was used to measure indoor temperature. Software programs to collect sensor data were also developed by the authors for the experiments, with each data being collected in real time at 10 s intervals. The installation of this sensor kit required a space that was approximately 20 cm in width and depth, and 15 cm in height. Each sensor costs approximately USD $350, which can be reduced to approximately $150 by minimizing the components to only measure the ${\mathrm{CO}}_{2}$ concentration. The sensor kits were all shipped to the research participants, who assembled and installed them independently following the instructions shown in Figure 3. Sensor kits were requested not to be installed on the floor or near heating and ventilation equipment.

The questionnaire was conducted once online in mid- to late-December 2020, asking participants to provide detailed answers about their housing attributes, building age, the number of occupants, household composition and heating methods, as well as questions about their awareness of ventilation and how often they ventilate their homes each season. Figure A1 in Appendix A shows the list of all questions of the questionnaire survey. In addition to these measurements and questionnaires, participants were approached separately if any additional details needed to be understood from their measurement data or questionnaire responses.

## 4. Results of Field Investigation

### 4.1. Measurement Results

#### 4.1.1. Effect of Combustion Heating Use

Figure 4 shows the monthly distribution of ${\mathrm{CO}}_{2}$ concentrations from October 2020 to March 2021 as a box plot. This box plot shows the maximum value excluding outliers, upper quartile, median, lower quartile, and minimum value excluding outliers, respectively, from top to bottom. In Figure 4, the vertical axis represents the ${\mathrm{CO}}_{2}$ concentration, while the horizontal axis represents the identification number, i.e., ID of each participant’s house. As shown in Table 2, houses from ID 15 onwards use combustion heaters.

The results shown in Figure 4 indicate that the ${\mathrm{CO}}_{2}$ concentration in many houses with combustion heating equipment (i.e., ID 15–24) had risen significantly since December 2020, reaching between 3500 ppm and 5000 ppm (in rare cases), as compared to houses without combustion heating. This result shows a similar trend to that of the existing surveys [9,10,11]. However, it cannot be concluded from this that the air is clean just because combustion heating is not used, as the concentration reaches around 3000 ppm in some houses (i.e., ID 1–14) that do not use combustion heating at all. This peculiarity is discussed in Section 5.

The aforementioned results also show that there are some houses with low ${\mathrm{CO}}_{2}$ concentrations, despite their use of combustion heating. This may be due to the use of other heating appliances such as air conditioners, although this balance varies from house to house. Figure 5 shows the ${\mathrm{CO}}_{2}$ concentration and indoor temperature on any given day for a house using both combustion heaters and air conditioners.

In both types of houses, a sharp increase in the ${\mathrm{CO}}_{2}$ concentration was observed at around 7:00 a.m., and the indoor temperature increased at the same time, which can be attributed to the use of combustion heating. At ID 17, as shown in Figure 5a, the ${\mathrm{CO}}_{2}$ concentration exceeded 3500 ppm in the morning and often reached 2000–3000 ppm during the day thereafter. In contrast, ID 24, as shown in Figure 5b, showed that the ${\mathrm{CO}}_{2}$ concentration reached 2500 ppm for a short period in the morning, but then remained in the range of 1000–2000 ppm. Based on indoor temperature trends, it can be estimated that the occupants of both houses are in the living room from morning to nighttime, or at least continue to use some kind of heating system in these rooms. Therefore, these results suggest that while IDs 17 and 24 use both combustion heating and air conditioning, the former uses combustion heating for a relatively long period of time, while the latter uses it for a short period of time. When participants were interviewed about their heating habits, it was found that occupants of ID 24 used combustion heating only for rapid heating when the indoor temperature was low, and air conditioning was the main heating source for the rest of the day. This difference in behavioral habits can be seen in the difference in concentration distribution after December, as shown in Figure 4.

#### 4.1.2. Relationship between ${\mathrm{CO}}_{2}$ Concentration and Outdoor Temperature

Figure 6 shows the daily averages of outdoor temperature and indoor ${\mathrm{CO}}_{2}$ concentration inside houses, values for which are plotted on a scatter plot. Each plot shows the fitting line, its equation, and the coefficient of determination $R^{2}$. Note that the range taken by the vertical axis depends on the concentration distribution of each house, in order to show the relative trend.

Results indicate that the outdoor temperature and ${\mathrm{CO}}_{2}$ concentration have negatively correlation in all houses except for ID 13, regardless of the heating appliance(s) used, although the strength of the correlation varies from house to house. As shown in Figure 4, the increase in ${\mathrm{CO}}_{2}$ concentration is more obvious in the winter season, when the outdoor temperature drops in houses using combustion heating; the same trend of increasing concentration can also be observed in houses that are not using combustion heating. This is especially noticeable in apartment houses ID 1–8, where the number of houses with concentrations reaching 2000 ppm increases as the outdoor temperature decreases.

The ${\mathrm{CO}}_{2}$ concentration trend for ID 14 is shown in Figure 7 and Figure 8 as examples of a house that can be assumed to have decreased ventilation under the influence of a decrease in outdoor temperature; gray periods in Figure 7 represent the missing measurements due to sensor failure. In this experiment, the missing rate of measurement was 1.38%. Since ID 14 does not use combustion heating, the ${\mathrm{CO}}_{2}$ concentration does not increase because of the use of a heating appliance. However, Figure 7 and Figure 8 show that the concentration increases from November to January. From this observation, it can be assumed that the ventilation frequency and volume gradually decreased as the outdoor temperature became colder.

Figure 9 and Figure 10 show the trend of ${\mathrm{CO}}_{2}$ concentration for ID 18, which uses combustion heating; the results shown in Figure 9 indicate that the indoor ${\mathrm{CO}}_{2}$ concentration in this residence increased rapidly after mid-December. ID 18 shows a similar gradual upward trend as ID 14 from October to November 2020, but the ${\mathrm{CO}}_{2}$ concentration begins to reach 3000 ppm at certain times of the day from mid-December onward, becoming higher still in January. The relationship between the outdoor temperature and indoor concentration shown in Figure 6, the difference is clear: ID 14 shows a linear relationship between outdoor temperature and concentration, although the $R^{2}$ value for ID 18 (0.633) is higher than the value for ID 14 (0.489), while ID 18 shows a non-linear change after an outdoor temperature of approximately 10 ${}^{{}^{\circ}}$C. The same trend is observed in IDs 16, 20, 22, etc., where combustion heating is used. The sudden increase in the concentration indicates that combustion heating begun to be used. From Figure 6 and Figure 10, it can be inferred that the timing of use in ID 18 is when the outdoor temperature drops to approximately 10 ${}^{{}^{\circ}}$C. On the other hand, ID 5, 10, and 21 apparently deviated from the negative correlation when the outside temperature was below 10 ${}^{{}^{\circ}}$C. In these houses, ventilation behavior, such as not opening windows to ventilate when it is cold outside, may have had an impact.

Thus, regardless of the type of heating equipment used, ${\mathrm{CO}}_{2}$ concentration tends to increase as the outdoor temperature decreases. The causes for this trend include combustion heating and a lack of ventilation.

For ID 13, the concentration distribution in January 2021 was lower than that in December 2020, as shown in Figure 4; the relationship between ${\mathrm{CO}}_{2}$ concentration and outdoor temperature for ID13 differed from that for the other houses, with a weak positive correlation, as shown in Figure 6. As for the former, it can be gathered from the results of the interviews that the door to the adjacent room was opened because the air conditioner broke down, and that the ventilation fan was cleaned in early January 2021, both of which may have affected the results. For the latter, the interviews revealed that pets were often in the living room and the air conditioner was always in use, except on days when the outdoor temperature was high. Owing to these circumstances, ID 13 showed a different trend from the other houses.

### 4.2. Questionnaire Results

This section describes the results of the questionnaire survey of participants regarding their awareness of ventilation and people’s behavior regarding ventilation.

Q7 and Q8 in Table 4 show the results of questions about interest in ventilation and whether the home is ventilated. In Q7, 20 out of the 24 participants answered that they were interested in ventilation. The majority of the respondents in Q8 answered that their self-evaluation of ventilation was “well enough done" or “more or less done,” reflecting the high level of interest in ventilation indicated in Q7.

Results of the questionnaire also supported the notion that the ventilation frequency was influenced by seasonal variations. In Q9-10 of Table 1, the number of participants who answered “always” decreased from 7 in spring and autumn to 2 in summer and winter. Conversely, the number of respondents who answered “almost never” increased from two to eight. Even among the participants who responded that the frequency of ventilation had not decreased, most of them showed an increase in the ${\mathrm{CO}}_{2}$ concentration with a decrease in temperature, as shown in Figure 6, which suggests that the frequency of ventilation decreased.

Q11 in Table 4 are the results of a question regarding the frequency of cleaning 24-h ventilation and ventilation openings in bathrooms and toilets. More than half of the participants answered “not at all” or “not sure.” High interest in ventilation does not necessarily lead to proper management of ventilation equipment.

Table 5 shows the results of the question about reasons for wanting to ventilate, and Table 6 shows the results of the question about reasons for not needing to ventilate. “Want to bring in fresh air” was indicated as the most common reason given for wanting ventilation; on the other hand, the most common reason for not ventilating was “because it’s cold (or hot)”, which is consistent with the trend shown in Figure 6.

## 5. Discussion

The state of IAQ was investigated through the measurement of ${\mathrm{CO}}_{2}$ concentrations in houses and questionnaire surveys, and these findings are discussed in this section.

### 5.1. IAQ Pollution from Combustion Heating

The use of oil- and gas-fired heating appliances was found to cause significant increases in ${\mathrm{CO}}_{2}$ concentrations, leading to indoor air pollution with concentrations ranging from 3000–5000 ppm. This is similar to the results of the existing studies described in Section 2; such an environment is both uncomfortable and raises concerns about health effects due to long-term exposure.

In contrast, the ${\mathrm{CO}}_{2}$ concentration was relatively lower in houses with multiple heating appliances, such as air conditioners, than in houses with only combustion heating, suggesting the possibility of reducing air pollution by using multiple heating methods. Although it is necessary to consider other factors, such as the airtightness of the house and the amount of ventilation, this result implies that the combined use of air conditioners and other devices might lead to IAQ improvement.

All of the existing measurement case studies were conducted only for the use of combustion heating, and the results described in this paper are findings obtained by targeting houses with various heating methods.

### 5.2. Influence of Outdoor Temperature on Ventilation

As shown in Figure 4 and Figure 6, the ${\mathrm{CO}}_{2}$ concentration tends to increase with decreasing outdoor temperature in many houses. This result can be attributed to a decrease in the frequency of ventilation, including window opening, in order to avoid a drop in room temperature due to air exchange. This is supported by the results of the questionnaire responses described in Section 4.2 and by the existing study [48].

Various pollutants such as CO, ${\mathrm{NO}}_{x}$, ${\mathrm{SO}}_{2}$, and VOCs are likely to be retained in houses with high ${\mathrm{CO}}_{2}$ concentrations and insufficient ventilation, which may cause Sick Building Syndrome and respiratory diseases caused by long-term exposure [13,14,36,37,38,39]. In addition, even when considering the effects of ${\mathrm{CO}}_{2}$ alone, high concentrations have negative effects, such as reducing the ability to think, concentrate, and sleep [26,27,28].

Changes in the ${\mathrm{CO}}_{2}$ concentration are difficult for occupants to perceive. It is necessary to implement methods to maintain a clean IAQ without relying on human senses, such as constantly measuring the concentration and encouraging ventilation at appropriate times.

### 5.3. Gap between Occupants’ Awareness of Ventilation and the Actual IAQ

The results of the measurements in houses and the questionnaire survey conducted in this study showed that the actual IAQ does not match occupants’ awareness of ventilation. This could be attributed to the inability of occupants to perceive air pollution, as well as a lack of knowledge about the ventilation system of the house and the effects of combustion heating as barriers.

On the other hand, occupants are highly interested in ventilation in their homes, and there is considerable room for improvement in residential IAQ. If appropriate solutions for improvement can be presented to occupants based on the findings of this study, they will be able to understand the actual conditions in their homes, and can be expected to change their behavior to achieve a healthy and comfortable air environment.

## 6. Conclusions

In this study, the true condition of residential IAQ was investigated by measuring the ${\mathrm{CO}}_{2}$ concentration and a questionnaire survey. From the results obtained, the relationship between factors such as the heating method, outdoor temperature, and air pollution was clarified. The main findings and suggestions obtained in this study are as follows:The ${\mathrm{CO}}_{2}$ concentration increased significantly with the use of combustion heating equipment, resulting in air pollution in the house. This air pollution could be reduced by using non-combustion heating methods such as air conditioners.In many houses, mainly apartment complexes, it was found that the ${\mathrm{CO}}_{2}$ concentration tends to increase as the outdoor temperature decreases. This result is presumably due to the fact that the frequency of ventilation, such as window opening, decreases in order to avoid a decrease in indoor temperature in the winter season, when the outdoor temperature decreases.The actual IAQ in each house does not match occupants’ awareness of ventilation. In addition to the occupants’ difficulty in perceiving air pollution, the lack of knowledge about the ventilation system and the effects of combustion heating might be additional barrier.

Based on these results, methods to improve residential IAQ should be studied in future work. Specifically in addition to the ${\mathrm{CO}}_{2}$ concentration, substances such as CO, ${\mathrm{NO}}_{x}$, and VOCs should be measured to clarify the relationship between each substance, and to investigate which behaviors lead to deterioration of the IAQ. Methods to improve the IAQ can subsequently be developed through technologies such as information provision and automation systems. It is important to utilize these methods to raise awareness of IAQ and to inform occupants about the harmful effects of pollutants. Since ventilation causes a decrease in heating efficiency, energy conservation in heating use is also an important issue. Future studies should also include methods to achieve both healthy IAQ and high energy efficiency.

## Figures and Tables

**Figure 1 sensors-22-07331-f001:**
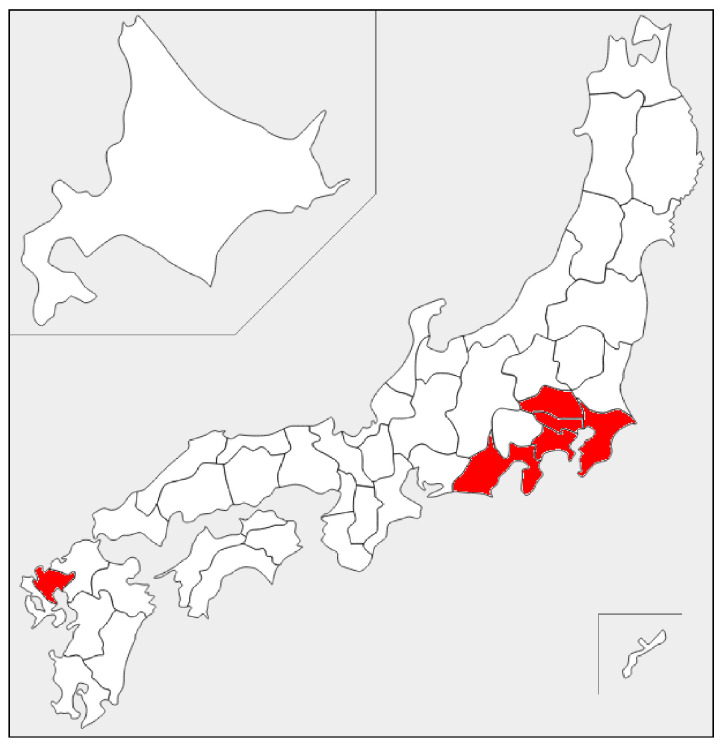
Target area of the experiments (in red). Extremely warm or cold regions were excluded.

**Figure 2 sensors-22-07331-f002:**
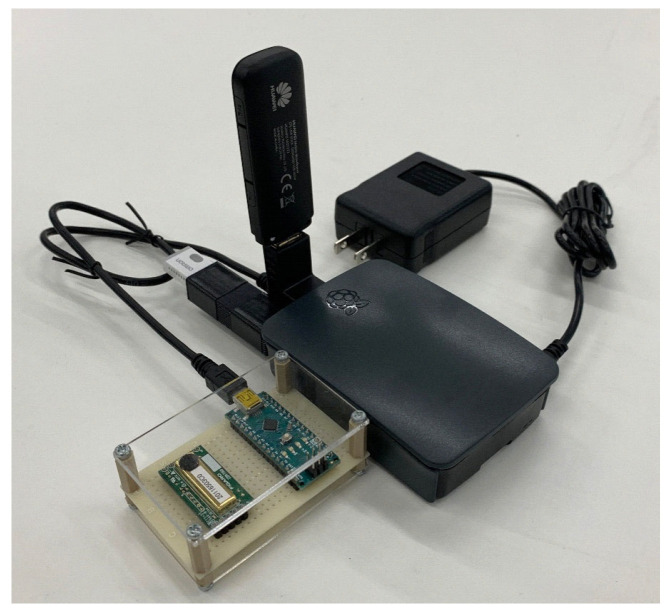
Sensor kit for measuring ${\mathrm{CO}}_{2}$ concentration.

**Figure 3 sensors-22-07331-f003:**
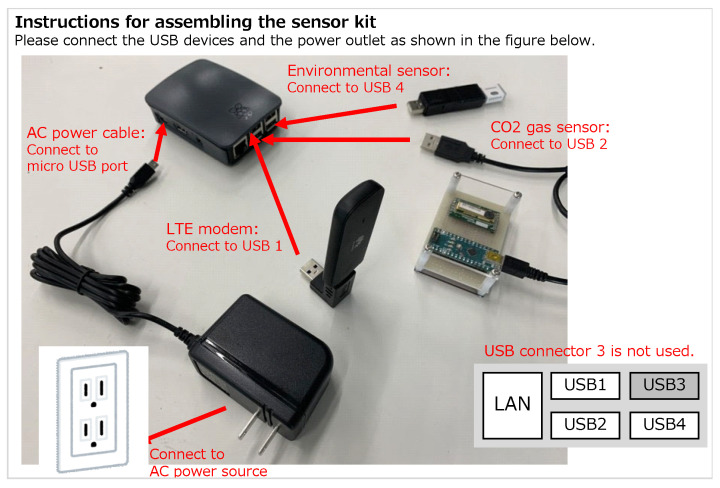
Instructions for assembling the sensor kit (distributed to research participants).

**Figure 4 sensors-22-07331-f004:**
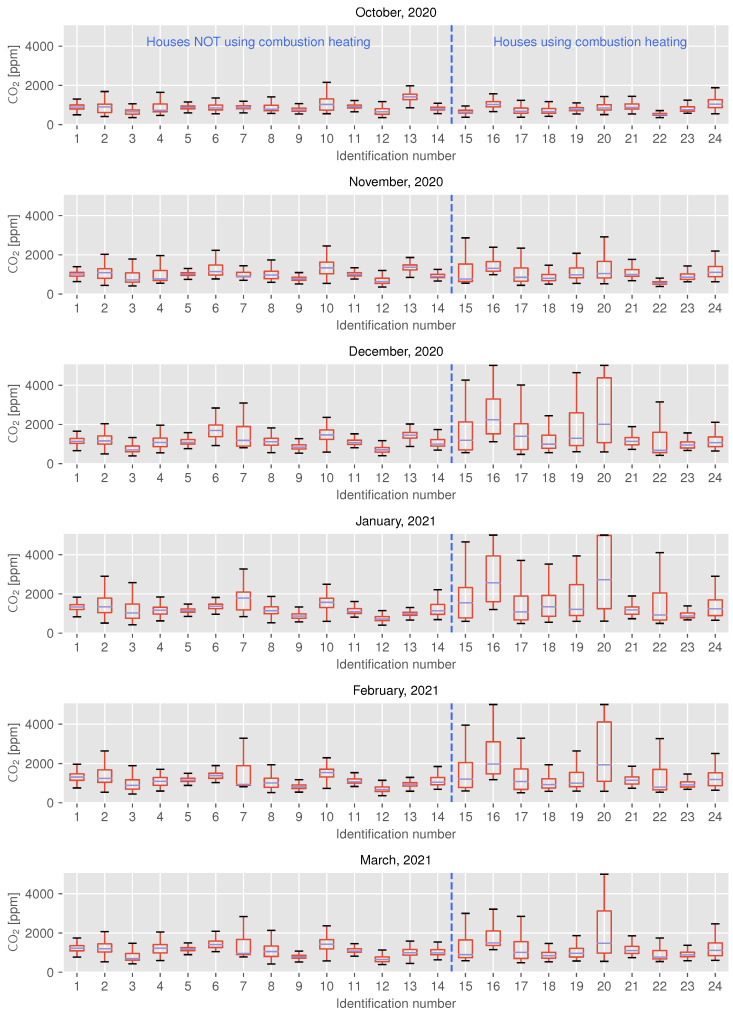
Monthly distribution of ${\mathrm{CO}}_{2}$ concentration in participants’ houses.

**Figure 5 sensors-22-07331-f005:**
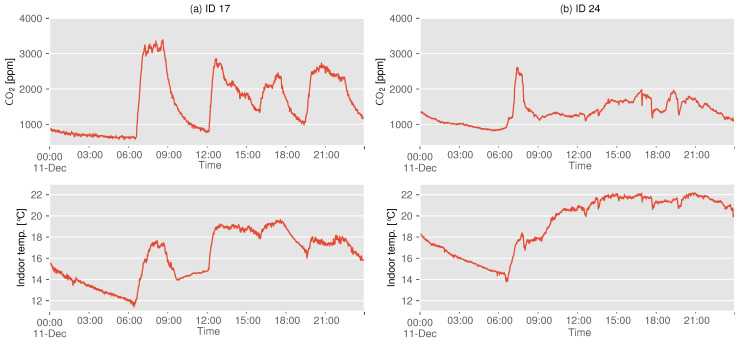
Indoor ${\mathrm{CO}}_{2}$ concentration and temperature when combustion heater is used at ID 17 and 24.

**Figure 6 sensors-22-07331-f006:**
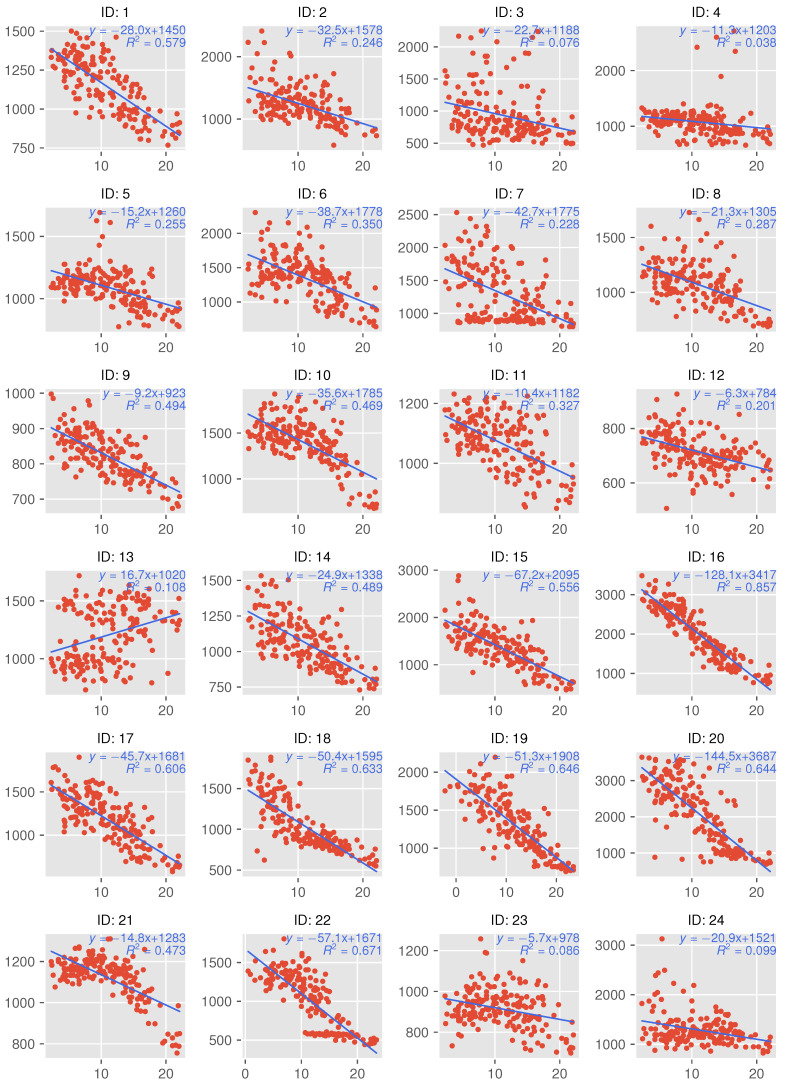
Relationship between daily averages of indoor ${\mathrm{CO}}_{2}$ concentration and outdoor temperature.

**Figure 7 sensors-22-07331-f007:**
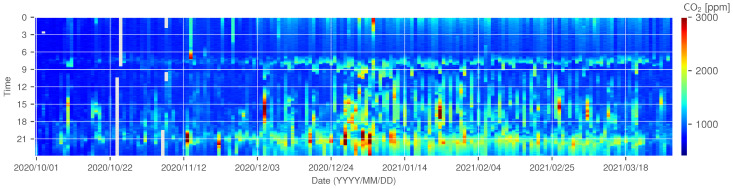
The heatmap of ${\mathrm{CO}}_{2}$ concentration at ID 14.

**Figure 8 sensors-22-07331-f008:**
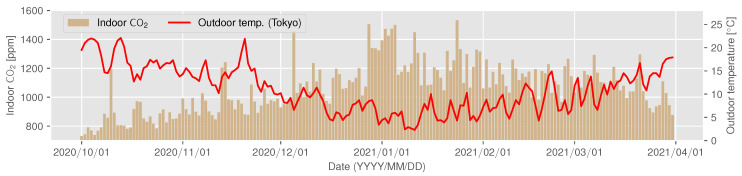
The changes in indoor ${\mathrm{CO}}_{2}$ concentration (vertical bars) and outdoor temperature (red line) at ID 14.

**Figure 9 sensors-22-07331-f009:**
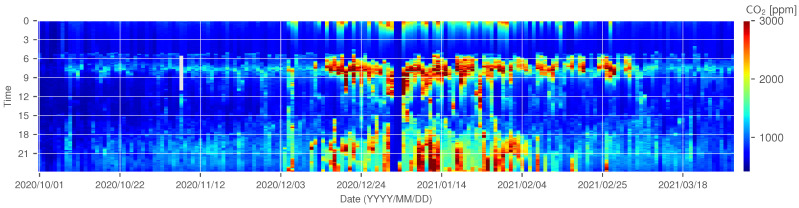
The heatmap of ${\mathrm{CO}}_{2}$ concentration at ID 18.

**Figure 10 sensors-22-07331-f010:**
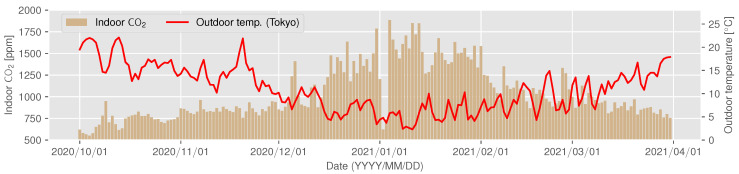
The changes in indoor ${\mathrm{CO}}_{2}$ concentration (vertical bars) and outdoor temperature (red line) at ID 18.

**Table 1 sensors-22-07331-t001:** ${\mathrm{CO}}_{2}$ concentration standards and guidelines in each country [18,22,23].

Country	Value	Target
Canada	1000 ppm	Residence
Finland	1200 ppm	Residence, office, school
France	1000 ppm	School and office
Germany	1500 ppm	School
Holland	1000–1500 ppm	Residence
Japan	1000 ppm	Buildings over a certain size
Korea	1000 ppm	Residence, office, school
New Zealand	1000 ppm	School
Norway	1000 ppm	School
United States	<700 ppm above outdoor level	Indoor

**Table 2 sensors-22-07331-t002:** Characteristics of research participants for the experiments.

ID	House	Structure	Year Built	# of Occupants	Heating Appliances in the Living Room
1	Apartment	RC	1999	3	Elec. heater
2	Apartment	RC	2003	5	AC (Air Conditioner), floor heating
3	Apartment	RC	2002	1	AC
4	Apartment	RC	2002	1	AC
5	Apartment	Steel-framed	1981	3	AC, elec. heating carpet
6	Apartment	RC	2000	2	AC, elec. heater
7	Apartment	RC	1986	1	AC, elec. heater
8	Apartment	Wooden	1966	1	AC
9	Detached	Wooden	2011	5	Floor heating
10	Detached	Wooden	2012	3	AC, floor heating
11	Detached	Wooden	2013	4	AC, kotatsu (table over an electric heater)
12	Detached	Wooden	1975	5	AC, Central AC
13	Detached	Wooden	2021	2	AC
14	Detached	Wooden	1989	4	AC, floor heating, elec. heater
15	Detached	Wooden	1987	2	**Gas heater**
16	Detached	Steel-framed	2014	2	**Gas heater**, kotatsu
17	Detached	Wooden	1990	4	**Gas heater**, AC
18	Detached	Wooden	2010	4	**Kerosene heater**, AC, heating carpet
19	Detached	Wooden	1964	2	**Kerosene heater**, AC, elec. heater, kotatsu
20	Detached	Wooden	1988	2	**Gas heater**
21	Detached	Wooden	1979	3	**Gas heater**, floor heating
22	Detached	Wooden	1975	3	**Kerosene heater**, AC, kotatsu
23	Detached	RC	2000	4	**Gas heater**, AC
24	Detached	Wooden	1997	3	**Kerosene heater**, AC, elec. heater

**Table 3 sensors-22-07331-t003:** Daily mean of outdoor temperature for Tokyo, Japan.

Day	2020			2021		
	Oct.	Nov.	Dec.	Jan.	Feb.	Mar.
1	19.5	14.3	10.2	4.4	7.2	11.8
2	21.0	15.6	8.2	4.8	8.8	13.5
3	21.7	14.8	8.1	3.7	6.2	7.4
4	22.0	13.7	9.2	5.8	7.0	9.0
5	21.7	13.4	6.6	6.0	7.1	11.8
6	21.0	12.8	8.2	5.3	8.9	13.8
7	18.3	15.5	9.7	6.3	9.9	7.8
8	14.7	17.3	11.4	2.4	6.1	6.6
9	14.5	13.5	9.8	2.9	4.7	9.6
10	16.0	11.9	9.3	2.6	6.7	12.0
11	19.8	11.9	10.2	2.3	8.7	9.9
12	21.5	10.2	11.4	3.4	7.8	12.7
13	22.1	13.7	9.9	5.8	10.4	11.1
14	20.3	14.7	8.5	7.9	12.3	12.5
15	16.8	13.4	6.1	6.6	11.4	12.9
16	16.1	14.7	4.6	10.0	10.6	14.5
17	12.7	15.2	4.3	6.1	7.6	13.7
18	14.3	15.8	6.1	4.3	4.3	12.3
19	13.1	19.1	5.8	4.4	6.9	12.9
20	15.6	21.9	4.4	3.9	10.2	14.1
21	16.0	16.6	5.3	4.8	13.9	16.7
22	17.4	15.0	5.6	8.7	14.9	12.1
23	16.8	15.5	7.7	6.4	11.0	10.8
24	17.3	11.8	8.0	4.3	6.4	13.8
25	15.5	10.5	8.7	7.6	6.6	14.5
26	16.2	12.8	6.4	7.5	8.7	14.5
27	16.7	11.6	7.8	10.3	5.7	13.7
28	16.6	11.8	8.5	4.4	6.6	16.5
29	17.3	10.1	8.7	5.3	-	17.5
30	14.9	9.9	6.9	4.1	-	17.8
31	13.8	-	3.4	5.4	-	17.9

**Table 4 sensors-22-07331-t004:** Responses to questions about ventilation concerns and behaviors. The darker the color of the answer, the worse it is for indoor air quality.

ID	Q7: Concernabout Ventilation	Q8: How Well Is Your House Ventilated?	Q9-10: Frequency of Ventilation		Q11: Frequency of Cleaning the Ventilation Filter
			Spring/Autumn	Summer/Winter	
1	Strongly interested	More or less done	Always	About once a day	About once every few months
2	Strongly interested	More or less done	Always	Almost never	Not at all
3	Not interested at all	Not done at all	Almost never	Almost never	Not sure
4	Strongly interested	Not much done	Almost never	Almost never	Not sure
5	Slightly interested	More or less done	About once every few hours	About once every few hours	Not sure
6	Slightly interested	Well done enough	About once every few hours	About once every few hours	Not at all
7	Strongly interested	More or less done	About once every few hours	About once every few hours	Not sure
8	Strongly interested	More or less done	At least once an hour	At least once an hour	About once every few months
9	Strongly interested	More or less done	Always	Almost never	About once every few months
10	Not very interested	More or less done	About once a day	About once a day	About once a year
11	Not very interested	Not much done	About once a day	Almost never	Not sure
12	Not very interested	More or less done	About once a day	Almost never	Not at all
13	Slightly interested	More or less done	About once every few hours	About once every few hours	About once a year
14	Strongly interested	Well done enough	Always	Always	Not at all
15	Strongly interested	Not much done	About once every few hours	About once a day	Not sure
16	Slightly interested	Well done enough	About once every few hours	About once every few hours	Not sure
17	Slightly interested	Well done enough	Always	Almost never	Not at all
18	Slightly interested	Well done enough	Always	About once every few hours	Not at all
19	Slightly interested	More or less done	Always	About once every few hours	Not at all
20	Slightly interested	More or less done	About once a day	About once a day	About once a year
21	Slightly interested	Not much done	About once every few hours	About once every few hours	Not at all
22	Slightly interested	Not much done	About once a day	Always	About once a year
23	Slightly interested	Not much done	About once a day	About once every few hours	About once a year
24	Strongly interested	Not much done	About once a day	Almost never	Not at all

**Table 5 sensors-22-07331-t005:** Answers to a question about the need for ventilation (multiple answers possible).

Q12: Why do you want to ventilate the room?	
**Answer**	**# of Responses**
To bring fresh air.	22
To adjust the indoor temperature.	5
It’s good for health.	3
To prevent colds and other infectious diseases.	7
To prevent condensation and mildew on windows.	8
The room smells bad.	11

**Table 6 sensors-22-07331-t006:** Answers to a question about when ventilation is not needed (multiple answers possible).

Q13: Why do you think it is not necessary to ventilate the room?	
**Answer**	**# of Responses**
Because the air outside is so polluted.	0
Because it’s cold (or hot) when ventilated.	19
Because it’ s already ventilated.	2
Because I’m using an air purifier.	1
Because some of us have hay fever.	4
Because some of us have respiratory problems.	0
I’ve never considered that I don’t need to ventilate.	4

## Data Availability

Not applicable.

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
