# Peer review of "Investigation of Indoor Air Quality in Residential Buildings by Measuring CO2 Concentration and a Questionnaire Survey"

_sensors, 2022, doi:10.3390/s22197331_

Round 1

Reviewer 1 Report

This paper entitles "Investigation of indoor air quality in residential buildings by measuring CO2 concentration and a questionnaire survey" and concerns the measurements of the air CO2 indoor for tracking the pollution of in the houses. Hattory and co-authors show indoor air quality (IAQ) in houses often deterioratea by chemical substances emitted from heating devices, expecially during cold periods.

The topic of this paper is relevant for Sensors MDPI. The objectives are presented in a clear way and both the strength and weaknesses of the method are described in the manuscript. Overall, the manuscript is well written, but needs some improvements before pubblication. My main concerns are in both the method section and the results sections. Some important information concerning the region of the experiment must be disclosed before pubblication. Some technical details of the sensor kit must be included in the manuscript. Some features of the questionnaires must be described with more details.

In conclusion, my general opinion is that this paper is publishable after minor revision.

I included my comments and suggestions in the annotated .pdf file.

Kind regards

Reviewer 2 Report

Review: Investigation of indoor air quality in residential buildings by measuring CO2 concentration and a questionnaire survey

In this paper, CO2 and temperature concentrations in indoor living rooms were measured by installing CO2 and temperature sensors in 24 Japanese households. In addition, a survey was conducted to the participants to confirm the level of awareness of indoor ventilation.

The contents of the review of the paper are as follows.

- In the introduction, an important point was identified that CO2 has a great effect on determining the degree of indoor ventilation and can be linked to the retention of hazardous substances.

- In the background, the allowable range of CO2 exposure in each country was explained and the relationship with other hazardous substances was also well explained.

- The study participants were relatively well set with 14 households that did not use combustion heating devices and 10 households that did.

In the IOT sensor part, more information about the sensor such as the sensor's accuracy and resolution is required.

- The results of the study clearly explained the difference between households that used combustion heating and those that did not.

- As a result of the survey, it was found that the actual figures and the perception of indoor ventilation did not match, and there was also a will to improve in the future.

As a result of reviewing this paper as a whole, the topic is consistent with the Atmosphere journal, and the research results are also relatively clearly explained, suggesting that publication is possible after minor revision.
